# Tri-Ponderal Mass Index: A Screening Tool for Risk of Central Fat Accumulation in Brazilian Preschool Children

**DOI:** 10.3390/medicina55090577

**Published:** 2019-09-08

**Authors:** Viviane Gabriela Nascimento, Ciro João Bertoli, Paulo Rogerio Gallo, Luiz Carlos de Abreu, Claudio Leone

**Affiliations:** 1Instituto Ciências da saúde da Universidade Paulista, UNIP, Curso de Nutrição, São Paulo 04043-200, Brazil; 2Departmento de Saúde, Ciclos de Vida e Sociedade, Faculdade de Saúde Pública da Universidade de São Paulo, São Paulo 0146-904, Brazil; 3Laboratório de Delineamento de Estudos e Escrita Científica, Centro Universitário Saúde ABC, Santo André 09060-870, Brazil; 4Programa de Mestrado em Políticas Públicas e Desenvolvimento Local da Escola Superior de Ciências da Santa Casa de Misericórdia, Vitória 29045-402, Brazil; 5Graduate Entry Medical School, University of Limerick, V94 T9PX Limerick, Ireland

**Keywords:** body mass index, tri-ponderal mass index, obesity, child, preschool, waist circumference, circumference–height ratio, central fat

## Abstract

*Background and Objectives*: To verify the use of the tri-ponderalmass index (TMI) as a screening tool for risk of central fat accumulation in preschool children. *Materials and Methods*: An observational, analytical study was carried out on samples from children 2 to 5 years of age. The body mass index (BMI) and the tri-ponderalmass index (TMI: Weight/height3) were calculated. The waist circumference-to-height ratio (WHtR) was used to classify central fat accumulation risk. Preschoolers whose WHtRwas in the upper tertile of the sample were classified as at risk for central fat accumulation. A comparison of the two indicators (BMI and TMI) was made from the area under the receiver operator characteristics (ROC) curve (AUC) in the discrimination of the WHtR. *Results*: The sample used for analysis was 919 preschoolers. The mean age of the children was 3.9 years (SD = 0.7). The difference in AUC was 5% higher for TMI (*p* < 0.0001). In the individual analysis of the ROC curve of the TMI, favoring a higher sensitivity, the cutoff point of 14.0 kg/m^3^ showed a sensitivity of 99.3% (95% CI: 97.6–99.9). *Conclusion*: Considering WHtR as a marker of possible future metabolic risk among preschool children, TMI proved to be a useful tool, superior to BMI, in screening for risk of central fat accumulation in preschool children.

## 1. Introduction

The prevalence of childhood obesity is still considered a significant public health problem because of its alarming progress. According to the World Health Organization (WHO), Latin America has made progress in the prevention and control of nutritional deficiencies but, on the other hand, is experiencing a rapid increase in the prevalence of overweight and obesity [1,2,3].

Recently, due to the association between excess body fat and the increased risk of developing coronary diseases, interest in quantifying the different compartments of the human body has increased too [4]. Currently, the body mass index (BMI), after many years of use, has been questioned because it presents limitations in the interpretation of its results as a screening tool [5,6].

Although BMI is recognized as an important prognostic indicator for some diseases, such as diabetes and cardiovascular disease, its adequacy as a phenotypic marker of adiposity has been contested, as it would not adequately estimate body composition, not differentiating fat mass and muscle mass, especially in children [6,7].

Initially proposed in the mid-1990s, the waist/height ratio (WHtR) has been related to several cardiovascular risk factors [8] and is considered a simple and effective indicator to measure abdominal obesity, indicating possible coronary risk in both adults and children [4].

According to Peterson et al., in children and adolescents, the tri-ponderal mass index (TMI), the ratio between body weight and height (kg/m^3^), estimates body fat excess more accurately than BMI, and they suggest replacing the use of BMI Z score by TMI [9]. 

Another study, even more recently, concluded that TMI would be a more accurate instrument to evaluate body fat, evidencing a good association with indicators of metabolic risk in the pediatric population [10]. TMI has been studied more in adolescents, and there are no references to preschool children.

Based on the hypothesis that TMI would also be useful in preschool children, the objective was to verify the use of the TMI as a screening instrument to identify the possible central accumulation of fat in children of this age group, which could be interpreted as an early marker of cardio-metabolic risk.

## 2. Methods

This is an observational, cross-sectional [11], and analytical study performed in a representative sample of children from 2.0 years to 4.9 years of age, enrolled in municipal daycare centers in the city of Taubaté, State of São Paulo, Brazil.

The city of Taubate is located in the interior of the state of São Paulo, 130 km from the city of São Paulo, state capital, and has 311,000 inhabitants (2018). It is the 53rd richest municipality in Brazil, has a high HDI-M 0.80 (UNDP 2017), is the 40th among 5570 municipalities in Brazil, and has public daycare centers and preschools with enough vacancies for the demand of children under 5 years of age.

Random sampling was by clusters, using the daycare centers a sample unit. Of the 59 municipal daycare centers existing in the list of the Secretary of Education and Culture of the Municipality of Taubaté, São Paulo, Brazil, nine were sequentially drawn, resulting in a final sample of 1284 children. The exclusion criteria were children with ages below 2.0 years or above 4.9 years, children with chronic diseases, congenital malformations, or specific growth diseases, children who did not attend daycare centers on the days selected for data collection, and children whose parents did not authorize participation.All other children enrolled in the selected daycare centers were initially included.

Of the initial total of 1284 preschool children, after elaboration and revision of the database, 365 children were excluded because they were outside the age range of the study (2.0 to 4.9 years of age) or presented inconsistency in their data, resulting in a final sample of 919 children.

The children were weighed on portable electronic scales (Seca^®^), with a capacity of 150 kg and precision of 100g, without shoes and with the minimum of clothes. For stature, we used a portable stadiometer (Wiso^®^), in cm and mm, fixed to the wall. In the measurement, the children placed their heels, calves, buttocks, and shoulders against the wall, positioning the head horizontally to the Frankfurt plane. All anthropometric measurements were obtained using the methods described by Lohman et al. [12].

The BMI was calculated from the measures of weight and height, and the TMI by the ratio of weight in kilograms to height in meters at the third power (TMI = W/H^3^). The risk classification of central fat accumulation was based on the ratio of waist circumference to height. 

For the comparison of the two markers, BMI and TMI, we used the respective areas under (AUC) the receiver operator characteristics (ROC) curve, obtained in the discrimination of children whose waist-to-height ratios were in the upper tertile of the sample, that is, classified as at high risk to central fat accumulation.

The Research Ethics Committee of the Faculty of Public Health of the University of São Paulo approved the present study (approval letter number 361/April 2009). The Informed Consent Form was sent to the mothers or guardians, with the support of the daycare center, being returned duly filled and signed before the beginning of the collection of the data collections.

## 3. Results

The sample analyzed includes 459 (49.9%) girls and 460 (50.1%) boys, distributed according to the following ages: 2.0 to 2.9 years, 57 (12.4%) girls and 64 (13.9%) boys; 3.0 to 3.9 years, 190 (41.4%) girls and 208 (45.2%) boys; and 4.0 to 4.9 years, 212 (46.2%) girls and 188 (40.9%) boys.

Table 1 shows that the analyzed parameters had very similar mean and median values.

As shown in Figure 1, there is a statistically significant (*p* < 0.0001) correlation between BMI and TMI, with a Pearson’s coefficient (*r*) of 0.83 (95% CI: 0.81 to 0.85). In the figure, there is a wide dispersion of data, in relation to the central tendency, in the higher values of TMI and BMI, which, however, is not statistically different to the coefficient presented by children with a TMI of less than 19 kg/m^3^ (*r* = 0.73 and *r* = 0.76, respectively; *p* = 0.6200).

Figure 2 shows differences between the ROC curves, where the AUC of BMI and TMI was 0.87 (95%CI: 0.85 to 0.89) and 0.92 (95%CI: 0.91 to 0.94), respectively. The observed difference between the areas, 0.05, was statistically significant (*p* < 0.0001).

Comparison of the AUC between TMI and BMI, by gender, showed no difference in relation to the group as a whole. AUC girls TMI = 0.93 (95% CI: 0.90–0.95) and BMI = 0.87 (95% CI: 0.84–0.91) and AUC boys TMI = 0.92 (95% CI: 0.89–0.94) and BMI 0.87 (95% CI: 0.83–0.90).

The comparison of the AUC of TMI by gender showed equal values, *p* = 0.5922 (not statistically significant).

In the isolated analysis of the ROC curve of TMI, taking into account the sensitivity and specificity, it can be observed that the cutoff point of 14.0 kg/m^3^ evinced high sensitivity (99.3 with 95%CI: 97.6 to 99.9), while the cutoff point of 17.0 had high specificity (94.5 with 95%CI: 92.4 to 96.6).

The Youden Index, corresponding to the TMI point that simultaneously optimizes the sensitivity and specificity values, respectively, 81.6 and 88.2, was 16.5 kg/m^3^ (Table 2).

## 4. Discussion

The preschool population studied presented very similar values of mean, median, and standard deviation, which characterizes a population with a near-normal distribution, i.e., almost without deviations to extreme values. 

Although there is a high correlation between BMI and TMI, in the analysis of its accuracy, through the ROC curves, the TMI shows a higher capacity for discrimination when used as an indicator of possible future risk of cardiometabolic disease. Several authors have advocated the use of ROC curves in order to define the accuracy of anthropometric measurements and the cutoff points for prediction of adiposity in children [13,14].

A systematic meta-analysis performed by Ashwell, Gunn, and Gibson [15] showed that the waist circumference-to-height ratio had a higher predictive capacity for metabolic risk and cardiovascular disease development than the isolated measure of waist circumference, this for both sexes. This review corroborates previous studies in which measures that reflected the central accumulation of adipose tissue were indicated to be markers of cardiometabolic risk, related to obesity, with a higher discrimination capacity than BMI or the absolute value of waist circumference [16,17].

Some authors suggest that values higher than 0.50 as a cutoff point may be a risk marker for the development of cardiovascular diseases in individuals of both sexes from six years of age, without mention of children under five years of age [18,19,20].

The term TMI was proposed by Peterson et al. [9] to differentiate from the Röhrer Index or Ponderal Index (PI) used to evaluate the intra-uterine growth, including the proportionality between the measures of length and body mass at birth. The PI is the ratio of mass in grams per length in centimeter^3^ and not of mass in kilograms per length in meters [3], like TMI. Although the PI was proposed many years ago, according to various authors, the literature about use of PI or TMI in the first years of life is restricted [9,10,21,22,23,24,25,26,27,28].

The TMI was proposed as a measure of body fat capable of reducing possible bias associated with the use of BMI, which, due to its variable values with age and gender, during growth implies the use of percentiles or z-scores to evaluate the results. The findings of the scientific literature indicate that there is a relative stability of TMI values during childhood and adolescence, making the use of age and sex specific percentiles practically unnecessary. As a result, TMI is a more precise and accurate tool for both body fat assessment and overweight status classification and is as simple to use and as accurate as the BMI percentiles [9,10,24,26].

Carrascosa et al. [26], to analyze reference values for BMI and TMI according to age in children without malnutrition or obesity, concluded that the TMI values could be used in clinical practice to screen for obesity in children and adolescents, instead of BMI z scores. 

Lorenzo et al. [25], with the objective of comparing BMI and TMI as predictors of fat mass percentage (FM%) and to develop TMI cutoff points, observed a high prevalence of adiposity in boys and girls. They suggested that TMI, instead of BMI, would be a better marker of the proportion of body fat mass for both sexes. According to these authors, TMI, when assessing adiposity in children and adolescents, showed a significantly larger area under the ROC curve than BMI. In this sense, the specific cutoff points for TMI classified adiposity better, allowing the conclusion to be made that TMI is a useful screening tool both in general pediatric clinical practice and in epidemiological studies on childhood obesity.

Despite its similarity to the Röhrer Index, TMI has not been studied as an indicator for nutritional assessment in the preschool age group, which makes it impossible to compare it to our research data. 

Our study corroborates the feasibility of using TMI as a valid instrument to screen children with obesity due to central fat accumulation, particularly in the primary care routine at preschool age.

Consequently, we analyzed different cutoff points for TMI. Of these, the cutoff point that simultaneously brought together the highest values of sensitivity and specificity (Youden J Index) corresponded to a TMI of 16.5 kg/m^3^, irrespective of the preschool children gender and age.

Considering primary care, if the problems arising from overweight already begin in preschool children, the goal of using TMI as a screening instrument for nutritional assessment should emphasize sensitivity, in order to enable the early start of care to prevent future metabolic and cardiovascular diseases.

In this sense, our results indicate that a better cutoff point for preschool screening, regardless of age or gender, would be a TMI of 14.0 kg/m^3^. Despite the low specificity, this value approaches a sensitivity of 100%. 

Now, these values need to be tested in new research in the primary health care of preschool children before proposing their use in the routine care of this age group.

## 5. Conclusions

As WHtR is a marker of possible future metabolic risk among preschool children because it represents the central accumulation of adiposity, TMI is a valid tool, superior to BMI, to the early identification of children with possible risk of cardiovascular and chronic non-communicable diseases in a primary health care setting.

## Figures and Tables

**Figure 1 medicina-55-00577-f001:**
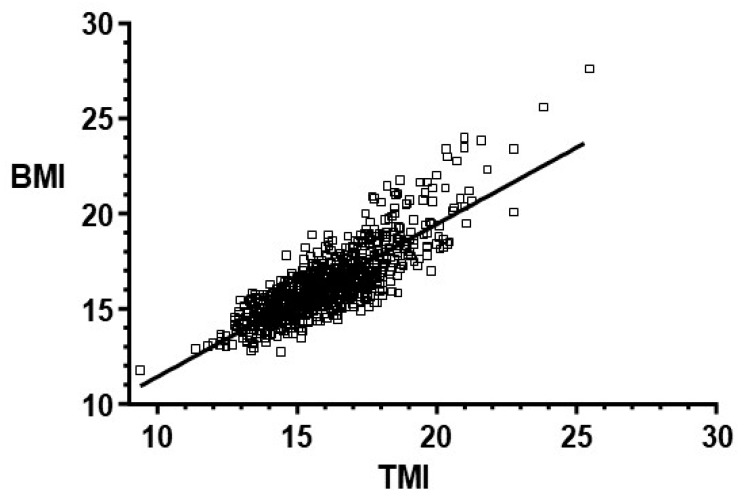
Linear regression between body mass index (BMI) and tri-ponderalmass index (TMI), preschool children. Taubaté, São Paulo, Brazil.

**Figure 2 medicina-55-00577-f002:**
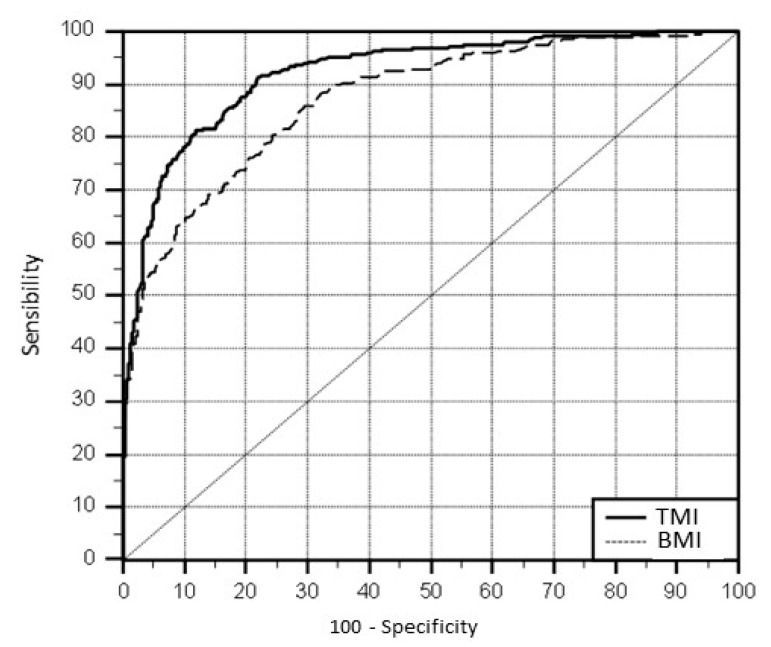
Comparison of the area under the curve (AUC) between body mass index (BMI) and tri-ponderalmass index (TMI) according to the risk of central fat accumulation, preschool children. Taubaté, São Paulo, Brasil.

**Table 1 medicina-55-00577-t001:** Median, mean, and standard deviation (SD) of weight, height, body mass index (BMI), waist circumference (WC), waist circumference-to-height ratio (WHtR), and tri-ponderal mass index (TMI) of preschool children. Taubaté, São Paulo, Brazil.

	Median	Mean	SD
Weight (kg)	16.4	16.9	3.2
Height (cm)	101.6	101.8	6.6
BMI	15.9	16.2	1.8
WC (cm)	51.5	52	5.1
WHtR	0.51	0.52	0.04
TMI	15.8	16	1.8

**Table 2 medicina-55-00577-t002:** Sensibility and specificity of cutoff points of the tri-ponderalmass index (TMI) according to receiver operator characteristics curve-to-screen risk of central fat excess in preschool-aged children.Taubaté, São Paulo, Brazil.

Cutoff Point (kg/m^3^)	Sensibility (CI 95%)	Specificity (CI 95%)
14	99.3 (97.6–99.9)	18.5 (15.5–21.8)
14.5	99.0 (97.1–99.8)	32.7 (29.0–36.6)
15	97.4 (94.8–98.8)	46.2 (42.2–50.2)
15.5	96.0 (93.2–97.9)	60.0 (56.0–63.9)
16	91.7 (88.0–94.6)	76.2 (72.6–79.5)
16.5 *	81.5 (76.6–85.7)	88.2 (84.8–90.2)
17	68.2 (62.6–73.4)	94.5 (92.4–96.5)

* Youden index J.

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
