# Peer review of "Tri-Ponderal Mass Index: A Screening Tool for Risk of Central Fat Accumulation in Brazilian Preschool Children"

_medicina, 2019, doi:10.3390/medicina55090577_

Round 1

Reviewer 1 Report

The authors presented and proposed the use of Tri-Ponderal Mass Index (TMI) as a better screening tool than traditional Body Mass Index (BMI) in the evaluation for the risk of central fat accumulation in preschool children. The waist circumference to height ratio (WHtR) was used to classify central fat accumulation risk. Considering WHtR as a marker of possible future metabolic risk among preschool children, TMI proved to be a useful tool, superior to BMI, in screening for risk of central fat accumulation in preschool children. However, the reviewer has some confusion. Major comments: The description about how the inclusion/exclusion criteria in this study were not mentioned clear enough in the text. The authors evaluated datasets of participants selected. General baseline characteristics and demographic data of the participants are encouraged to be provided as a separate table for clarification and comparison. Please clarify and provide the approval number for the conduction of the study by the Research Ethics Committee. Although descriptions about the limitations of the study, as well as the disadvantages of the applications of TMI in the evaluation of the metabolic conditions of the participants are missing, the authors presented the above-mentioned models in details. The authors analyzed and addressed the experimental data clearly, compared and discussed in-depth with the common state-of-the-art methods, et al., such as that of BMI. This study found the useful basic information for further studies in the field.

Author Response

Dear Reviewer
I submit the requested modifications.

Regards

PS: There is an article with modifications by both reviewers entitled: "Medicina-549585 Revis 1 e 2" in this manuscript the modifications of the opinion 1 are in red while the changes corresponding to the opinion 2 are in green.

Reviewer 2 Report

This is an interesting paper but must be improved in certain aspects

The title must be more concrete, indicating that these are Brazilian preschoolers.

The bibliography does not respect the regulations. sometimes it is cited according to the Vancouver protocol (1, 2) and other times with the author's name and the year in brackets: Peterson et al (2017).

If the acronym TMI is used for the triponderal index, throughout the text it should always be cited as TMI and not rarely with the acronym and other times with the full name.

Eliminate the entire paragraph between line 60 an line 67. Simply, explain what were de "N" of the sample and how many were girls and boys. Explain how the perimeter of the waist was measured, as well as the mathematical expression of waist to height ratio and BMI. In the material and methods section explains why a correlation analysis was performed and what the Youden index was used for.

Describe the variability of direct dimensions and indexes by sex, contrasting whether there are significant differences. In the event that sex diferences exists, the ROC analysis should be done separately.

line 106: what is IMR ?

the conclusion are  very "risky" TMI is a valid tool, superior to BMI, to early identify children with central adiposity, eventualy related with cardiovascular diseases, but this aspect is not derived of your work Predicting percentage body fat through waist-to-height ratio (WtHR) in Spanish schoolchildren.

It is recommended to broaden the discussion and to incorporate the following bibliography:

Marrodán et al. Predicting percentage body fat through waist-to-height ratio (WtHR) in Spanish schoolchildren.Public Health Nutrition 17, 870-876 (2014).

Mesa et al. Anthropometric parameters in screening for excess of adiposity in Argentinian and Spanish adolescents: evaluation using receiver operating characteristic (ROC) methodology. Annals of Human Biology 40, 396-405 (2013)

Author Response

Dear Reviewer
Submitting requested changes

Regards

PS: There is an article with modifications by both reviewers entitled: "Medicina-549585 Revis 1 e 2" in this manuscript the modifications of the opinion 1 are in red while the changes corresponding to the opinion 2 are in green.

Round 2

Reviewer 2 Report

The authors have satisfactorily resolved the comments and suggestions.
Congratulations on your work.